# Intestinal-Type Adenocarcinoma Is a Rare Histotype of Vulvar Neoplasm: Systematic Review of the Literature

**DOI:** 10.3390/cancers17243989

**Published:** 2025-12-14

**Authors:** Alessio Colalillo, Dominga Boccia, Luigi Della Corte, Daniele Neola, Federica Rosato, Silvia D’Ippolito, Maria De Ninno, Damiano Arciuolo, Maurizio Guida, Giuseppe Bifulco, Francesco Cosentino

**Affiliations:** 1Division of Gynecologic Oncology, Department of Woman and Child Health and Public Health, Fondazione Policlinico Universitario A. Gemelli IRCCS, Università Cattolica del Sacro Cuore, 00168 Rome, Italy; colalillo1987@gmail.com; 2Department of Neuroscience, Reproductive Sciences and Dentistry, School of Medicine, University of Naples Federico II, 80138 Naples, Italy; dominga.boccia@gmail.com (D.B.); luigi.dellacorte@unina.it (L.D.C.); maurizio.guida@unina.it (M.G.); 3Responsible Research Hospital, 86100 Campobasso, Italy; f.rosato3@studenti.unimol.it (F.R.); maria.deninno@responsible.hospital (M.D.N.); damiano.arciuolo@policlinicogemelli.it (D.A.); francesco.cosentino@unimol.it (F.C.); 4Department of Medicine and Health Science “V. Tiberio”, University of Molise, 86100 Campobasso, Italy; silvia.dippolito@unimol.it; 5Department of Public Health, School of Medicine, University of Naples Federico II, 80138 Naples, Italy; giuseppe.bifulco@unina.it

**Keywords:** intestinal-type adenocarcinoma, primary vulvar adenocarcinoma, vulvar cancer

## Abstract

Intestinal-type vulvar adenocarcinoma (VAIt) is an exceptionally rare variant of primary vulvar adenocarcinoma, characterized by a mucinous, villo-glandular architecture and an intestinal immunophenotype. To define its clinicopathological features, management, and outcomes, a systematic review was conducted. A comprehensive search of five databases (1998–2025) identified 40 VAIt cases, including one from our institution. Patients ranged from 31 to 92 years (median 58), with tumors most frequently arising on the labia. Diagnosis relied on recognition of intestinal-type morphology and confirmatory immunohistochemistry, particularly CK20 and CDX2 positivity, while excluding metastatic colorectal cancer. Most tumors were early-stage. Surgical treatment—most commonly wide local excision—was the primary management strategy, with heterogeneous use of adjuvant radiotherapy or chemotherapy. Outcomes were generally favorable, with 72.5% of patients disease-free at follow-up, though recurrences and disease-related deaths underscore the tumor’s unpredictable behavior and need for long-term surveillance. Given the rarity of VAIt, collaborative studies and molecular characterization are essential to develop standardized diagnostic and treatment guidelines.

## 1. Introduction

Vulvar cancer is a rare disease, accounting for about 4–6% of all gynecologic cancers. It most commonly affects postmenopausal women with a mean age at diagnosis of 68 years, but the incidence is constantly increasing. The main risk factors are HPV infection (especially HPV-16 and HPV-18), smoking habit, obesity, immunosuppression, poor genital hygiene, and chronic vulvar dermatoses such as lichen sclerosus [1,2,3,4,5,6,7,8,9,10,11,12,13,14,15,16,17,18,19,20,21,22,23,24,25,26,27,28,29,30,31,32,33]. There are several histological types of vulvar cancer; however, the most frequent (about 90%) is squamous cell carcinoma (SCC). Other histologic types, such as melanomas, sarcomas, basal cell carcinomas, and adenocarcinomas, are rarer. Among these rare entities, primary vulvar adenocarcinoma (VA) represents an uncommon diagnosis. Due to its low prevalence, information regarding VA clinical progression and long-term outcomes is scarce [34]. Vulvar adenocarcinomas often originate from Bartholin glands and are most diagnosed in women aged 50 to 60 years. In several cases, their clinical presentation closely resembles benign conditions, such as Bartholin duct cysts. Occasionally, these tumors can arise from other vulvar structures including minor vestibular glands and Skene’s glands, or they may be associated with heterotopic mammary tissue, endometriosis, or even coexist with Paget’s disease [35]. A particularly unusual subtype of VA is Intestinal-type Vulvar Adenocarcinoma (VAIt), also referred to in older literature as “Cloacogenic Carcinoma”—although such terminology is no longer supported by the most recent WHO classification (5th edition, 2020) [36]. According to the WHO, VAIt is defined as a mucinous, villo-glandular adenocarcinoma of the vulva showing intestinal differentiation. Macroscopically, VAIt may exhibit a polypoid appearance, while histologically, it displays features akin to mucinous colorectal adenocarcinomas, including the presence of goblet cells or Paneth cells with abundant intracellular mucin. A definitive diagnosis requires not only the recognition of this intestinal-type morphology, but also a confirmation through immunohistochemical profiling, such as CK20 and CDX2 [37]. The diagnosis must be substantiated by the exclusion of other primary tumors, in particular primary gastrointestinal neoplasias. Due to its rarity, the development of therapeutic guidelines for VAIt is controversial and there is no standardized treatment described. The aim of this study was to provide a comprehensive synthesis of the existing literature on VAIt, also reporting a case of VAIt from our institution, in order to define its clinical, pathological, immunohistochemical characteristic, management, and prognosis.

## 2. Materials and Methods

### 2.1. Search Strategy

The entire study protocol was registered prospectively in the PROSPERO International register of systematic reviews (CRD420251090699). The Preferred Reporting Item for Systematic Reviews and Meta-analyses (PRISMA) statement and checklist were adopted for reporting the systematic review of literature [38]. All review stages, including search strategy, study selection, risk of bias assessment, data extraction, and data analysis, were independently performed by two authors (D.B and A.C). Disagreements were resolved by consensus with a third reviewer (F.R).

A structured literature review was conducted across MEDLINE, EMBASE, Web of Science, SCOPUS, and Cochrane Library databases, using a combination of the following MESH terms: “Intestinal-type adenocarcinoma”; “Primary vulvar adenocarcinoma”; “Vulvar cancer” from 1 January 1998—when VAIt was first described by Tiltman and Knutzen [2]—to 31 May 2025.

### 2.2. Study Selection

Only articles written in English were included. Inclusion criteria consisted of clinical reports or studies of any other design specifically describing cases of VAIt. References list from each eligible study were also screened for missed studies. In vitro or animal models studies, proceedings of scientific meeting or abstracts, and studies describing tumors with mixed histological features or arising outside of the vulva or external genitalia (e.g., involving the vagina) were excluded from the analysis.

### 2.3. Data Extraction

Data extraction was performed without modification of the original data. For each included study, we collected the following data: study type, year, country, tumor location, tumor size (i.e., maximum diameter), FIGO stage, presence of lymphovascular space invasion (LVSI), presence of lymph node metastasis, immunohistochemical (IHC) features, clinical management (i.e., surgical management and/or adjuvant treatment), survival outcomes (i.e., progression-free survival), and follow-up information.

### 2.4. Risk of Bias Assessment

To assess the risk of bias within the included studies, an adapted version of the Joanna Briggs Institute (JBI) critical appraisal tool for case reports was utilized [39]. This assessment, focusing on reporting quality, does not represent a formal risk of bias evaluation but rather highlights potential limitations in the available evidence. The following criteria were used:Patient Characteristics: the reporting of patient demographics, medical history, clinical presentation, and comorbidities. Scores were assigned as follows: 2 = Comprehensive description; 1 = Basic description; 0 = Minimal/absent information.Condition/Diagnosis: The description of the condition, diagnostic process, differential diagnoses, and diagnostic investigations. Scores were assigned as follows: 2 = Detailed; 1 = Adequate; 0 = Limited.Intervention: the reporting of the intervention (e.g., surgery, chemotherapy), including details on the procedure, dosage, timing, and modifications. Scores were assigned as follows: 2 = Detailed; 1 = Adequate; 0 = Limited.Outcomes: the reporting of primary and secondary outcomes, including complications and adverse events. Scores were assigned as follows: 2 = Detailed; 1 = Adequate; 0 = Limited.Follow-up: the duration and completeness of follow-up. Scores were assigned as follows: 2 = Extended follow-up; 1 = Short-term follow-up; 0 = None/Unclear.Diagnostic Tests: the description of diagnostic tests used for diagnosis confirmation. Scores were assigned as follows: 2 = Detailed description and rationale; 1 = Key tests mentioned; 0 = Limited/None.

A total reporting quality score was then calculated for each study by summing the scores across the six criteria, with a maximum score of 12. A higher score indicates a more complete reporting.

## 3. Results

### 3.1. Study Selection

After the database searches, 9614 studies were identified. Duplicate removal led to 887 articles. The abstract screening process led to the selection of 50 selected papers to evaluate in full text. Based on the evaluation of the full text, we included 32 papers in the qualitative analysis [2,3,4,5,6,7,8,9,10,11,12,13,14,15,16,17,18,19,20,21,22,23,24,25,26,27,28,29,30,31,32,33] (Figure 1), with a total of 40 cases of VAIt (including our case, reported in Appendix A).

### 3.2. Studies and Patients’ Characteristics

The 32 included studies consisted of individual case reports and small case series [2,3,4,5,6,7,8,9,10,11,12,13,14,15,16,17,18,19,20,21,22,23,24,25,26,27,28,29,30,31,32,33]. Specifically, 27 of these references (84.4%) presented individual case reports focusing on a single patient, while the remaining 5 references (15.6%) comprised either case series or grouped individual case reports from various publications, collectively contributing data from multiple patients.

In total, 40 cases of VAIt (including our case) were found. The characteristics of the included studies are reported in Table 1 and Table 2.

Regarding patient demographics, the age at diagnosis demonstrated a broad spectrum, ranging from 31 to 92 years, with a calculated median age of 58 years (based on available data points). Cases originated from diverse geographical locations, including the USA, Japan, Italy, Sweden, China, Korea, Turkey, France, and Spain. Tumor location was frequently reported in the labium (left, right, or unspecified) in 12 cases (30%), followed by the fourchette in 4 cases (10%), the vestibulum in 3 cases (7.5%), and the periurethral region in 3 cases (7.5%). Other reported sites included the vulva (unspecified) in seven cases (17.5%), the Bartholin gland in two cases (5%), the hymen in two cases (5%), the posterior commissure in one case (2.5%), between the introitus and anus in one case (2.5%), and the distal lower third of the right hemivulva in one case (2.5%). An analysis of tumor characteristics revealed primary tumor diameters varying from 7 mm to 60 mm. LVSI status was infrequently documented: out of the 40 cases, only 7 (17.5%) were explicitly stated as negative for LVSI, while in 32 cases (80%) this parameter was not reported. Lymph node metastasis status, however, was available for 32 cases (80%). Among these, 22 cases (68.8% of those with known status) were reported as negative for lymph node metastasis, while 10 cases (31.2% of those with known status) were positive. For eight cases (20% of total cases), lymph node status was not specified. Immunohistochemical staining results consistently pointed towards an intestinal phenotype. IHC characteristics of the included cases are reported in Table 3. When reported, CEA was positive in 16/17 cases (94%), CK7 in 17/28 cases (60,7%), CDK20 in 26/28 cases (93%), and CDX2 in 18/18 cases (100%). When reported, CK20 immunostaining was greater than CK7 in 16/18 cases (89%). None of the extracted cases were HPV-correlated.

The examination of FIGO staging among the 40 reported cases offered a clearer understanding of disease extent at diagnosis. Stage IA emerged as the most frequently reported classification, encompassing 19 cases (47.5%); 6 cases were categorized as Stage IB, (15%); 2 cases (5%) were noted as Stage IIB; 4 cases were classified as Stage IIIA (10%); Stage IIIB was identified in 1 case (2.5%), and Stage IIIC in 2 cases (5%). Beyond these, one case (2.5%) presented as the more advanced Stage IVB. A single instance of carcinoma in situ was also documented, representing 2.5% of cases. Lastly, for four cases (10%), the FIGO stage was either not applicable or not explicitly specified.

Regarding clinical management, surgical resection was the most common treatment for these patients. Wide local excision alone was the approach in 11 cases (27.5%). More extensive surgical interventions included radical vulvectomy combined with lymph node dissection in 11 cases (27.5%), and wide local excision with lymph node or sentinel lymph node dissection, also in 7 cases (17.5%). Other specific surgical procedures reported in individual cases included local excision with ipsilateral lymph node dissection in one case, (2.5%), radical hemicolectomy in one case, (2.5%), laparoscopic posterior pelvic exenteration with radical vulvectomy and bilateral inguinal lymph node dissection in one case (2.5%), radical vulvectomy with bilateral inguinal lymph node dissection in one case (2.5%), radical vulvectomy with bilateral inguinal lymphadenectomy, rectal resection, and reconstruction in one case (2.5%), salpingo-oophorectomy with systematic pelvic lymphadenectomy, radical vulvectomy, and radical colpectomy in one case (2.5%), and partial vulvectomy in one case (2.5%). Surgical treatment was not reported in two cases (5%). Adjuvant therapy was administered in a notable proportion of cases; 23 cases (57.5%) received no adjuvant treatment. Radiotherapy (RT) was given in four cases (10%), chemotherapy (CT) in three cases (7.5%), and a combination of chemotherapy and radiotherapy (CT + RT) in three cases (7.5%). For seven cases (17.5%), adjuvant therapy status was not specified. In one specific case, at diagnosis, the treatment consisted of exclusive radiotherapy, while upon recurrence, surgical intervention was performed, followed by adjuvant chemoradiotherapy. Ultimately, an examination of outcomes revealed wide variability in progression-free survival (PFS), ranging from 4.5 months to 120 months. The most favorable outcome, “No disease” (indicating no evidence of recurrence at follow-up), was reported in 29 cases (72.5%). Four cases (10%) resulted in death due to the disease, and relapse was noted in one case (2.5%). The outcome was not specified for six cases (15%).

### 3.3. Risk of Bias Assessment

The quality of the reporting assessment of the included studies is presented in Appendix A. Across the 32 included studies, the overall reporting quality was moderate, with most studies providing adequate, but often not detailed, descriptions of patient characteristics, diagnostic assessment, intervention, outcomes, and follow-up. Total quality scores ranged from 5 to 7, with most reports scoring 6, indicating consistent adherence to core reporting elements. The main area of variability was the reporting of diagnostic tests, where several earlier publications offered limited or no information, resulting in lower domain scores. In contrast, more recent reports, particularly those published from 2022 onward, demonstrated more comprehensive diagnostic workups and often included molecular analyses, reflected in higher scores for this domain.

## 4. Discussion

Intestinal-type vulvar adenocarcinoma (VAIt) represents an exceptionally rare and poorly understood primary neoplasm of the vulva. Our systematic review, including 41 documented cases (including the present one), synthesized existing knowledge on its clinicopathological features, therapeutic approaches, and outcomes, providing a valuable resource for future clinical practice guidelines. Given the extremely low incidence of VAIt, individual case reports and small case series represent the only source of information about this condition in the literature, which inherently limits the power of definitive conclusions and uniform treatment protocols, underscoring the limited understanding of its pathogenesis, clinical behavior, and optimal management strategies [40]. Due to its uncommon occurrence, diagnosis can be challenging and typically requires thorough immunohistochemical profiling to differentiate it from metastatic colorectal carcinoma [41].

Vulvar cancer itself is a rare malignancy, globally accounting for approximately 0.65% of all female cancers [34]. According to recent GLOBOCAN estimates, there were an estimated 52,192 new cases of vulvar cancer and 17,998 deaths worldwide in 2022 [42]. Uncommon histological variants of vulvar carcinoma, including adenocarcinomas, comprise less than 5% of all vulvar cancer diagnoses worldwide. VAIt is described as a sporadic variant of vulvar carcinoma [42]. The demographic profile of VAIt patients in our review indicates a broad age range at diagnosis, from 31 to 92 years, with a median age of 58 years. This aligns with previous observations suggesting its prevalence in women during their fifth to sixth decade. VAIt most commonly affects postmenopausal women and typically presents as a solitary lesion in the perianal or posterior vulvar area [4]. The specific origin of VAIt remains a subject of ongoing debate. Hypotheses suggest derivation from cloacal remnants persisting in adults or from intestinal metaplasia of Müllerian or skin adnexal structures [12]. According to the ontogenetic theory proposed by Novak [1], these tumors are thought to originate from misplaced embryonic intestinal tissue or from cloacal remnants that persist in the adult urogenital tract. Tiltman and Knutzen [2] proposed that aberrant embryonic tissue in the vulva could potentially develop into a type of intestinal-like adenocarcinoma. This theory posits that primitive cloacal epithelium, which has the potential for intestinal differentiation, can give rise to these unique adenocarcinomas in locations like the vulva. The cloaca, which initially forms the lower vagina, urethra, and rectum during embryonic development, might leave behind fragments of gastrointestinal-type tissue. These remnants, particularly in the vestibular area of the vulva, may persist into adulthood and later transform into malignant tumors resembling those seen in the colon. Rodriguez et al. [8] introduced the term “neometaplasia” to describe the occurrence of well-differentiated tissues in tumors found in locations with no embryological connection to those tissue types. It is often localized to epithelial glands in the vulvar region, frequently associated with Bartholin’s glands, but can also arise from minor vestibular glands, Skene’s glands, endometriosis implants, or aberrant mammary tissue [6]. Microscopically, VAIt is characterized by a villo-glandular structure composed of goblet cells or Paneth cells, exhibiting intracytoplasmic mucin similar to those found in colorectal adenocarcinomas. This structural resemblance is further supported by the consistent presence of an intestinal-type immunohistochemical phenotype, characterized by high rates of CK20 positivity, CDX2 positivity, and CEA positivity, often accompanied by CK7 positivity [12]. The imperative for thorough diagnostic and radiological examination is paramount to definitively exclude other primary malignancies, thereby confirming the vulvar origin. Unfortunately, the included studies did not uniformly report which diagnostic tests were perform to exclude a primary gastrointestinal adenocarcinoma in each included case, representing a limitation of our review.

Analysis of tumor characteristics revealed primary tumor diameters spanning 7 mm to 60 mm. The status of LVSI was infrequently documented in the reviewed literature. This scarcity of information on LVSI limits our ability to determine its precise prognostic significance in VAIt [12].

In our results, lymph node metastasis status was available for 32 cases (80%). Among these, 10 cases (31.2%) were reported as having positive lymph nodes. This finding is particularly notable as some studies suggest a higher rate of positive inguinal nodes in early-stage VAIt compared to vulvar squamous cell carcinomas. The prevalence of positive lymph nodes, even in early-stage disease, underscores the importance of regional lymph node assessment in VAIt management [26,36]. One case of lymph node staging performed with sentinel lymph node technique has been reported, presumably performed on the basis of indications for sentinel node procedure as per the GROINSS-V study recommendations [43]. Regarding the disease stage, direct correlation between FIGO stage and outcome could not be definitively established for every individual case in this review due to the limited number of detailed follow-up data for each specific stage; the general principles of cancer staging strongly suggest that higher FIGO stages in VAIt would correlate with poorer progression-free survival and overall survival, necessitating more aggressive therapeutic approaches [44]. The occurrence of “death due to the disease” in 10% of our reviewed cases, and relapses in 2.5%, often associated with more advanced stages or aggressive tumor biology, further supports the prognostic significance of FIGO staging in this rare variant. Given the rarity of VAIt, standardized treatment guidelines are yet to be definitively established. However, based on the accumulated evidence, surgical intervention remains the cornerstone of treatment for removable disease [44]. Our review highlights that wide local excision was the predominant surgical approach, employed in 27.5% of cases, aiming to achieve clear margins. More extensive resections, such as radical vulvectomy combined with lymph node dissection (27.5%) or wide local excision with lymph node or sentinel lymph node dissection (17.5%), were also common, often dictated by tumor size, location, and local invasion. Lymph node assessment, whether via sentinel node biopsy or complete dissection, is consistently recommended, particularly for tumors exceeding 2 cm or when imaging suggests lymphatic involvement [45]. This is a critical step given the observed rate of positive lymph nodes (31.2% of cases with known status) in our cohort, reflecting the potential for regional spread. The role of neoadjuvant therapy in VAIt is currently unclear, with limited reported experiences. While some advanced-stage (FIGO Stage III) cases have explored neoadjuvant chemotherapy followed by radical surgery as a potential strategy, more data are needed to ascertain its efficacy and indications. Similarly, the use of adjuvant therapy is not yet standardized [44]. Our review reveals significant heterogeneity in adjuvant treatment decisions, with a substantial proportion of cases (57.5%) not receiving any post-surgical therapy. When administered, radiotherapy (RT) was given in 10% of cases, chemotherapy (CT) in 7.5%, and combined chemoradiotherapy (CT + RT) in 7.5%. For 17.5% of cases, adjuvant therapy status was not specified. This variability underscores the lack of consensus on optimal adjuvant strategies, often leading to individualized approaches based on tumor characteristics, surgical margins, lymph node status, and physician preference [37]. Adjuvant chemotherapy, often utilizing fluoropyrimidine or platinum-based regimens, and radiotherapy have been reported to show some success in advanced or recurrent cases. The documented case in our review, in which exclusive radiotherapy was given at diagnosis and then surgical intervention followed by adjuvant chemoradiotherapy upon recurrence, strikingly exemplifies the adaptive and evolving nature of treatment strategies when faced with disease progression in such a rare malignancy [46]. This highlights the urgent need for collaborative efforts to establish more uniform treatment protocols through larger multi-institutional studies or centralized registries. Ultimately, an examination of clinical outcomes revealed considerable variability in progression-free survival (PFS), ranging from 4.5 months to 120 months. Encouragingly, “No disease” (indicating no evidence of recurrence at follow-up), was the reported outcome for the majority of patients (72.5%). However, VAIt can follow an unpredictable clinical course, with potential for late recurrence and metastasis [40]. Death was documented in four cases. Relapse was found in only one case (2.5%) described by Voltaggio et al. [23]. The unpredictable behavior of VAIt, including the potential for relapses even many years after initial treatment, underscores the critical need for extended follow-up periods to gain a comprehensive understanding of optimal therapeutic management and long-term prognosis [47].

Recent whole exome sequencing (WES) analyses of HPV-independent VAIt cases have provided valuable insights into the genomic alterations of this rare tumor type. These studies have consistently identified pathogenic TP53 mutations and CD274/PD-L1 amplification in VAIt. While sharing commonalities like clock-like mutational signatures (SBS1 and SBS5) associated with aging and DNA damage, these WES analyses also revealed distinct copy number alterations among cases, suggesting biological heterogeneity. The presence of TP53 abnormalities suggests that emerging p53-based therapies may represent potential treatment targets for VAIt [48,49].

Moreover, genomic and transcriptomic analyses provided evidence that VAIt may share closer molecular characteristics with colorectal adenocarcinomas than with Müllerian-origin carcinomas. This similarity opens the exciting possibility of exploring the efficacy of systemic drugs typically prescribed for colorectal cancers in the treatment of VAIt [12]. In particular, the therapeutic landscape of colorectal adenocarcinoma has been transformed by the introduction of immunotherapy, especially in tumors exhibiting microsatellite instability (MSI-H) or mismatch repair deficiency (dMMR) [50]. If comparable molecular features were identified in VAIt, immune checkpoint inhibitors, such as PD-1 or PD-L1 blockers, could represent a rational treatment option. Although evidence remains limited, these parallels may justify further investigation of immunotherapy as part of the systemic treatment options for VAIt, with the potential to improve outcomes in a malignancy for which standardized therapeutic protocols are lacking.

Consequently, comprehensive histopathological and molecular characterization is increasingly important for guiding personalized treatment decisions in this exceedingly rare and challenging malignancy [34]. However, due to the rarity of this type of neoplasm, more case reports, molecular studies, and centralized registries are needed to develop standardized diagnostic and treatment guidelines. More experiences and longer follow-up periods are needed to elucidate the best therapeutic management and its long-term prognosis.

## 5. Conclusions

Our findings highlight the significant challenges in managing VAIt due to its rarity and lack of established guidelines. The optimal treatment is surgical in most cases; however, it should be highly individualized and tailored to the patient’s specific clinical history and tumor characteristics. The recurrence in 5.4% of cases, including the present case, underscores the need for vigilant long-term follow-up. The observed lymph node positivity in early-stage disease also emphasizes the importance of comprehensive staging.

Further studies, particularly multicenter collaborations, centralized registries, and molecular analyses, are crucial to deepen our understanding of VAIt. This will enable the development of standardized diagnostic criteria, refined therapeutic protocols, and a clearer long-term prognostic understanding for this exceptionally rare vulvar neoplasm.

## Figures and Tables

**Figure 1 cancers-17-03989-f001:**
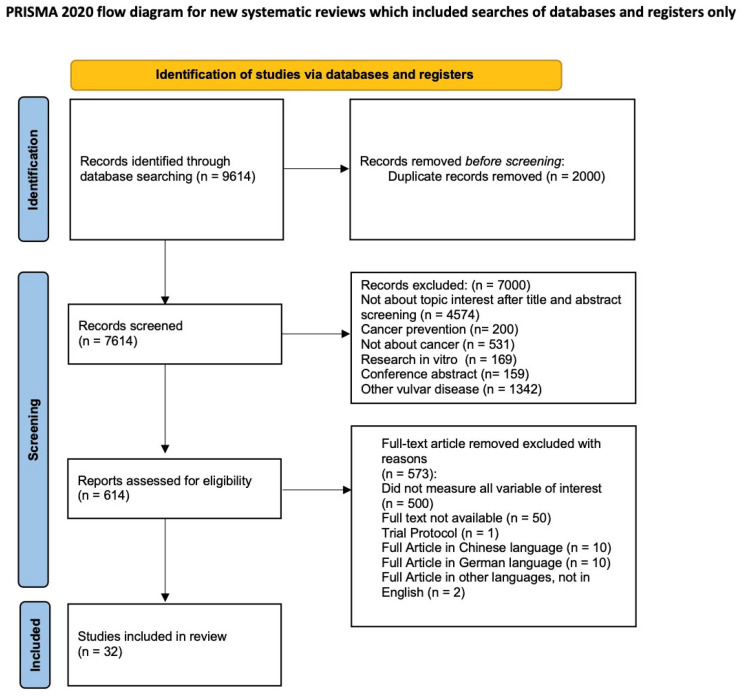
PRISMA 2020 flow diagram.

**Table 1 cancers-17-03989-t001:** Characteristics of the included studies.

Reference (Year)	Country/Region	Age (Year)	Location	Diameter of Tumor (mm)	LVSI (%) [Negative- Positive]	Lymph Node Metastasis [Negative-Positive]
Tiltman et al. (1978) [2]	NS	50	Periurethral	20	NS	positive
Kennedy et al. (1993) [3]	NS	54	Left posterior	20	NS	negative
		63	Fourchette	15	NS	negative
Ghamande et al. (1995) [4]	Boston, Massachusetts	67	Vulva	12	NS	negative
Willen et al. (1999) [5]	Göteborg, Sweden	57	Vestibulum	10	NS	negative
Ohno et al. (2001) [6]	Chiba, Japan.	92	Bartholin gland	50	NS	positive
Zaidi et al. (2001) [7]	Birmingham, USA	43	Fourchette	50	NS	negative
Rodriguez et al. (2001) [8]	Granada, Spain	69	Right labium	15	NS	negative
Liu et al. (2003) [9]	Taipei, Taiwan	49	Left labium	18	negative	negative
Dube et al. (2004) [10]	Québec, Canada.	58	Right labium	15	negative	negative
Dube et al. (2006) [11]	Québec, Canada.	64	Hymen	7	NS	NS
Cormio et al. (2012) [12]	Bari, Italy	59	Vestibulum	NS	NS	positive
		42	Vulva	10	NS	negative
Karkouche et al. (2012) [13]	Paris, France	31	Fourchette	NS	NS	NS
Musella et al. (2013) [14]	Rome, Italy	57	Right labium	50	NS	negative
Sui et al. (2016) [15]	Shaanxi, China	43	Hymen	15	NS	negative
Tulek et al. (2016) [16]	Ankara, Turkey	62	Vulva	30	NS	positive
Matsuzaki et al. (2017) [17]	Okinawa, Japan	68	Periurethral/Vestibulum	40	NS	NS
He et al. (2017) [18]	Beijing, China	63	Vulva	20	NS	NS
Lee et al. (2017) [19]	Seoul, Korea	64	Right labium	50	negative	NS
Tepeoglu et al. (2018) [20]	Ankara, Turkey	40	Left labium	20	NS	positive
Kurita et al. (2019) [21]	Kitakyushu, Japan	63	Periurethra	20	NS	negative
Kaltenecker et al. (2019) [22]	Lillington, USA	53	Left labium	60	NS	positive
Voltaggio et al. (2020) [23]	Baltimore, USA	43	NS	30	NS	NS
		38	NS	NS	NS	NS
Robinson et al. (2021) [24]	Oslo, Norway	55	Bartholin Gland	NS	NS	negative
Laforga et al. (2021) [25]	Alicante, Spain	45	Left labium	30	NS	negative
Martin-Vallejo et al. (2021) [26]	Alicante, Spain	45	Left labium	30	NS	negative
Moscoso et al. (2022) [27]	Barcelona, Spain	66	Left labium/Fourchette	22	negative	negative
Sato et al. (2022) [28]	Osaka, Japan	63	Vulva	10	negative	negative
Sopracordevole et al. (2023) [29]	Aviano, Italy	63	Right labium	10	negative	positive
Mateoiu et al. (2024) [30]	Gothenburg, Sweden	63	Between introitus and anus	30	negative	NS
		63	Posterior commissure	20		negative
Natsume et al. (2024) [31]	Tokyo, Japan	63	Left labium minus	30	NS	positive
Trecourt et al. (2025) [32]	Lyon, France	55 (mean) *n* = 4 cases	Vulva	29 (mean)	NS	negative
Fujii et al. (2025) [33]	Tokyo, Japan	58	Vulva	30	NS	positive
Present case (2025)	Campobasso, Italy	58	Distal lower third of the right hemivulva	30	positive	positive

Four studies have two cases (Kennedy et al. [3], Cosmio et al. [12], Voltaggio et al. [23], Mateoiu et al. [30]) and one has four cases (Trecourt et al. [32]). NS: not specified.

**Table 2 cancers-17-03989-t002:** Type of treatment, stage, and outcomes.

Reference (Year)	FIGO Stage	Treatment Modalities	Adjuvant Therapy	Progression-Free Survival (Months)	Outcomes
Tiltman et al. (1978) [2]	IA	modified radical vulvectomy + LND	No	12	No disease
Kennedy et al. (1993) [3]	IA	radical vulvectomy + LND	No	120	No disease
	IA	wide local excision	No	48	No disease
Ghamande et al. (1995) [4]	IA	radical vulvectomy + LND	No	17	No disease
Willen et al. (1999) [5]	IIIA	wide local excision	No	26	No disease
Ohno et al. (2001) [6]	IA	No	RT	10	Died of disease
Zaidi et al. (2001) [7]	IA	modified radical vulvectomy + LND	No	18	No disease
Rodriguez et al. (2001) [8]	IB	wide local excision	No	36	No disease
Liu et al. (2003) [9]	IIB	wide local excision + LND	No	24	No disease
Dube et al. (2004) [10]	IA	radical hemicolectomy	No	16	No disease
Dube et al. (2006) [11]	IA	wide local excision	No	4.5	NS
Cormio et al. (2012) [12]	IVB	radical vulvectomy + LND	No	54	Died of disease
	IA	radical vulvectomy + LND	No	39	No disease
Karkouche et al. (2012) [13]	IA	wide local excision	No	15	No disease
Musella et al. (2013) [14]	III	wide local excision + ipsilateral LND	No	17	No disease
Sui et al. (2016) [15]	IA	wide local excision	CT	24	No disease
Tulek et al. (2016) [16]	IVB	wide local excision	CT	36	Died of disease
Matsuzaki et al. (2017) [17]	Carcinoma in situ	wide local excision	No	60	No disease
He et al. (2017) [18]	IB	wide local excision	No	26	No disease
Lee et al. (2017) [19]	IB	wide local excision	No	12	No disease
Tepeoglu et al. (2018) [20]	IIIA	partial vulvectomy	No	38	No disease
Kurita et al. (2019) [21]	IB	wide local excision + LND	RT	12	No disease
Kaltenecker et al. (2019) [22]	IA	no	CT + RT	12	Died of disease
Voltaggio et al. (2020) [23]	NA	no	NS	NS	NS
	NA	no	NS	72	Relapse
Robinson et al. (2021) [24]	IB	wide local excision + SLN	NS	NS	No disease
Laforga et al. (2021) [25]	IA	wide local excision + SLN	No	6	No disease
Martin-Vallejo et al. (2021) [26]	II	wide local excision + LND	No	8	No disease
Moscoso et al. (2022) [27]	IA	wide local excision + LND	No	12	No disease
Sato et al. (2022) [28]	IB	wide local excision	No	27	No disease
Sopracordevole et al. (2023) [29]	IIIC	wide local excision + LND	CT + RT	20	No disease
Mateoiu et al. (2024) [30]	IA	modified radical vulvectomy	No	64	No disease
	IA	radical vulvectomy + LND	RT	17	No disease
Natsume et al. (2024) [31]	IIIC	laparoscopic posterior pelvic exenteration with radical vulvectomy and bilateral inguinal lymph node dissection	RT	6	No disease
Trecourt et al. (2025) [32]	IA (3 cases)-IIB (1 case)	surgical resection **	Ns	Ns	Ns
Fujii et al. (2025) [33]	IIIA	radical vulvectomy, bilateral inguinal lymph node dissection	RT	17	No disease
Present case * (2025)	IIIB	exclusive RT-CT (at diagnosis)/radical hysterectomy type B2 with bilateral salpingo-oophorectomy and systematic pelvic lymphadenectomy, radical vulvectomy, radical colpectomy (recurrence treatment)	CT + RT	48	Relapse
			CT	2	No disease

LND: lymph node dissection; NS: not specified; CT: chemotherapy; RT: radiotherapy; SLN: sentinel lymph node biopsy; * present case has a first diagnosis in 2021 with relapse, and second diagnosis in 2025; ** type of surgery not specified.

**Table 3 cancers-17-03989-t003:** Immunohistochemical characteristics of the included studies.

Reference (Year)	CEA	P53	ER/PR	CDX2	CK20	CK7	CK20/CK7 Ratio	P16 (Pattern)	HPV Correlation
Tiltman et al. (1978) [2]	NA	NA	NA	NA	NA	NA	NA	NA	NA
Kennedy et al. (1993) [3]	−	NA	NA	NA	NA	NA	NA	NA	NA
Ghamande et al. (1995) [4]	+	NA	NA	NA	NA	NA	NA	NA	NA
Willen et al. (1999) [5]	+	NA	NA	NA	NA	NA	NA	NA	NA
Ohno et al. (2001) [6]	+	NA	NA	NA	NA	NA	NA	NA	NA
Zaidi et al. (2001) [7]	+	+	−/−	NA	NA	NA	NA	NA	NA
Rodriguez et al. (2001) [8]	+	NA	−/−	NA	+	+	NA	NA	NA
Liu et al. (2003) [9]	NA	NA	NA	NA	NA	NA	NA	NA	NA
Dube et al. (2004) [10]	NA	NA	−/−	NA	+	+	NA	NA	NA
Dube et al. (2006) [11]	NA	NA	NA	NA	-	+	NA	NA	No
Cormio et al. (2012) [12]	NA	NA	NA	NA	NA	+	NA	NA	No
	NA	NA	NA	NA	+	+	NA	NA	No
Karkouche et al. (2012) [13]	NA	NA	NA	NA	+	−	CK20 strongly + CK7−	NA	No
Musella et al. (2013) [14]	+	NA	ER -	+	+	−	CK20 strongly + CK7 focally +	+ (strongly)	No
Sui et al. (2016) [15]	NA	NA	NA	NA	−	+	CK7 diffusely + CK20−	+ (focally)	No
Tulek et al. (2016) [16]	NA	NA	NA	+	+	+	CK20 strongly + CK7 focally +	NA	No
Matsuzaki et al. (2017) [17]	NA	NA	NA	+	+	−	CK20 strongly + CK7−	NA	No
He et al. (2017) [18]	+	NA	ER -	+	+	+	CK20 strongly + CK7 focally +	−	No
Lee at al. (2017) [19]	+	+	NA	+	+	+	CK20 strongly + CK7 strongly +	+ (focally)	No
Tepeoglu et al. (2018) [20]	+	NA	NA	+	+	+	CK20 strongly + CK7 focally +	NA	No
Kurita et al. (2019) [21]	NA	NA	NA	NA	+	−	CK20 strongly + CK7−	NA	No
Kaltenecker et al. (2019) [22]	+	+	NA	NA	+	−	NA	NA	NA
Voltaggio et al. (2020) [23]	NA	NA	NA	NA	+	+	NA	+ (block type)	No
	NA	NA	NA	NA	+	+	NA	+ (block type)	No
Robinson et al. (2021) [24]	+	NA	NA	+	+	+	CK20 strongly + CK7 strongly +	NA	NA
Laforga et al. (2021) [25]	NA	NA	NA	+	+	−	NA	−	No
Martin-Vallejo et al. (2021) [26]	NA	NA	NA	+	+	−	CK20 strongly + CK7−	−	No
Moscoso et al. (2022) [27]	+	NA	NA	+	+	−	CK20 focally + CK7−	+	NA
Sato et al. (2022) [28]	NA	NA	−/−	+	+	+	CK20 strongly + CK7 focally +	NA	No
Sopracordevole et al. (2023) [29]	+	+	−/−	+	+	+	CK20 strongly + CK7 focally +	NA	NA
Mateoiu et al. (2024) [30]	+	−	−/+	+	+	+	NA	+ (patchy)	No
	NA	+	NA	+	+	NA	NA	+ (patchy)	No
Natsume et al. (2024) [31]	+	NA	NA	+	+	+	CK20 strongly + CK7 focally +	+ (patchy)	No
Trecourt et al. (2025) [32]	NA	NA	NA	+	+	−	CK20 strongly + CK7−	NA	No
Fujii et al. (2025) [33]	NA	NA	ER -	+	+	−	CK20 strongly + CK7−	+ (patchy)	No
Present case (2025)	+	NA	−/−	+	+	−	CK20 strongly + CK7−	+ (patchy)	No

NA: not assessed.

## Data Availability

The data that support the findings of this study are available upon reasonable request from the corresponding author.

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
