# Peer review of "Intestinal-Type Adenocarcinoma Is a Rare Histotype of Vulvar Neoplasm: Systematic Review of the Literature"

_cancers, 2025, doi:10.3390/cancers17243989_

Round 1

Reviewer 1 Report

Comments and Suggestions for Authors

In this article, the authors summarize the clinicopathologic and immunohistochemical characteristics, management, and prognosis of intestinal-type vulvar adenocarcinoma (VAIt), a rare histotype of vulvar neoplasm. Overall, this is a comprehensive systematic review, and the manuscript is well written. However, I do have some comments and suggestions:

  1. In practice, the diagnosis of VAIt is challenging for both surgical and gynecologic pathologists. The lack of a centralized pathology review in this type of literature analysis raises concerns regarding diagnostic accuracy. For this reason, the authors should make a greater effort to address this limitation of the systematic review. For example, CK7 and CK20 are very useful markers in the diagnosis of VAIt; however, they cannot simply be reported as “+” or “–.” Instead, the relative intensity of these markers should be compared, and if CK7 < CK20, this pattern supports a lower-gastrointestinal immunophenotype. Similarly, when reporting p16 positivity, is a “block-type” pattern observed? Is there any possibility of HPV-related lesions?
  2. Table 3 summarizes the “Reporting Quality Assessment of Included Case Reports.” This lengthy table provides minimal meaningful information and should be moved to the supplementary materials.
  3. In Table 3, the authors state that “Total reporting quality scores ranged from 5 to 11.” Is this correct? It appears that the maximum possible score is 7.
  4. Figure S3: Panels A, B, and C are mislabeled in the figure legend. If possible, I also recommend moving this figure into the main manuscript rather than leaving it in the supplementary section.

Author Response

REVIEWER 1

Comment #0

  1. A) In this article, the authors summarize the clinicopathologic and immunohistochemical characteristics, management, and prognosis of intestinal-type vulvar adenocarcinoma (VAIt), a rare histotype of vulvar neoplasm. Overall, this is a comprehensive systematic review, and the manuscript is well written. However, I do have some comments and suggestions.
  2. B) Response: We thank the Reviewer for the kind comment.
  3. C) Location: \

Comment #1

  1. A) In practice, the diagnosis of VAIt is challenging for both surgical and gynecologic pathologists. The lack of a centralized pathology review in this type of literature analysis raises concerns regarding diagnostic accuracy. For this reason, the authors should make a greater effort to address this limitation of the systematic review. For example, CK7 and CK20 are very useful markers in the diagnosis of VAIt; however, they cannot simply be reported as “+” or “–.” Instead, the relative intensity of these markers should be compared, and if CK7 < CK20, this pattern supports a lower-gastrointestinal immunophenotype. Similarly, when reporting p16 positivity, is a “block-type” pattern observed? Is there any possibility of HPV-related lesions?
  2. B) Response: We thank the Reviewer for the clever comment. We agree that CK7 and CK20 could be useful markers in the diagnosis of VAIt. Therefore we assessed which of the included studies compared the intensity of CK7 and CK20, as well as which studies reported a “block type” pattern of p16 positivity and HPV relation. We included such information in the revised manuscript.
  3. C) Location: Table 3

Comment #2

  1. A) Table 3 summarizes the “Reporting Quality Assessment of Included Case Reports.” This lengthy table provides minimal meaningful information and should be moved to the supplementary materials.
  2. B) Response: We thank the Reviewer for the comment. We agree and we moved Table 3 to supplementary material (Supplementary Table 1) to increase readability of the manuscript.
  3. C) Location: Supplementary Table 1.

Comment #3

  1. A) In Table 3, the authors state that “Total reporting quality scores ranged from 5 to 11.” Is this correct? It appears that the maximum possible score is 7.
  2. B) Response: We thank the Reviewer for the comment. We checked the text and in fact the maximum score in the included studies is 7, as reported in the table. We corrected the text accordingly.
  3. C) Location: Lines 212-213

Comment #4

  1. A) Figure S3: Panels A, B, and C are mislabeled in the figure legend. If possible, I also recommend moving this figure into the main manuscript rather than leaving it in the supplementary section.
  2. B) Response: We thank the Reviewer for the comment. We checked and modified the labels of Figure S3 to increase clarity. We did not move this figure to the main manuscript, as we designed this study as a systematic review and we decided to include the information about the case from Our Institution in Supplementary Material in accordance with the Editorial Board of the Journal. However, we are available to move such information to the main text if the Reviewer and the Editor decide to.
  3. C) Location: Supplementary Figure 3.

Reviewer 2 Report

Comments and Suggestions for Authors

The manuscript “Intestinal-type adenocarcinoma is a rare histotype of vulvar neoplasm: systematic review of literature” is a well-conducted and valuable contribution to the limited body of knowledge on this exceptionally rare tumor. The authors have performed a rigorous systematic review, prospectively registered on PROSPERO, and complemented it with a well-documented institutional case. The study is clearly structured, adheres to PRISMA standards, and offers a comprehensive synthesis of clinicopathologic, immunohistochemical, and molecular findings.

The review’s strengths lie in its methodological rigor, the breadth of data collected (41 documented cases), and the inclusion of recent genomic insights that enrich the discussion. The manuscript is well written and of direct relevance to gynecologic oncologists and pathologists.Only minor revisions are suggested to enhance clarity and presentation. A graphical summary of key findings (for instance, stage distribution, immunoprofile, and outcomes) would improve readability. The time frame of the search (“to May 2025”) should be clarified to avoid ambiguity. Figures could benefit from clearer labeling, and a brief expansion of the discussion on the diagnostic and therapeutic implications of molecular profiling (including NGS and MMR testing) would add further depth.Overall, this is an excellent and comprehensive review that deserves publication after minor revisions. The authors are to be commended for their meticulous work on such a rare and challenging entity.

Author Response

REVIEWER 2

Comment #1

  1. A) The manuscript “Intestinal-type adenocarcinoma is a rare histotype of vulvar neoplasm: systematic review of literature”is a well-conducted and valuable contribution to the limited body of knowledge on this exceptionally rare tumor. The authors have performed a rigorous systematic review, prospectively registered on PROSPERO, and complemented it with a well-documented institutional case. The study is clearly structured, adheres to PRISMA standards, and offers a comprehensive synthesis of clinicopathologic, immunohistochemical, and molecular findings.

The review’s strengths lie in its methodological rigor, the breadth of data collected (41 documented cases), and the inclusion of recent genomic insights that enrich the discussion. The manuscript is well written and of direct relevance to gynecologic oncologists and pathologists. Only minor revisions are suggested to enhance clarity and presentation. A graphical summary of key findings (for instance, stage distribution, immunoprofile, and outcomes) would improve readability. The time frame of the search (“to May 2025”) should be clarified to avoid ambiguity. Figures could benefit from clearer labeling, and a brief expansion of the discussion on the diagnostic and therapeutic implications of molecular profiling (including NGS and MMR testing) would add further depth.Overall, this is an excellent and comprehensive review that deserves publication after minor revisions. The authors are to be commended for their meticulous work on such a rare and challenging entity.

  1. B) Response: We thank the Reviewer for the comment. We clarified the exact date of search and improved the tables and figures to increase clarity.
  2. C) Location: Lines 109-110; Tables and Figures

Reviewer 3 Report

Comments and Suggestions for Authors

This manuscript addresses an exceptionally rare entity, intestinal-type vulvar adenocarcinoma (VAIt), by combining a systematic literature review with a single institutional case. The topic is clinically relevant, the rarity justifies aggregation, and the synthesis could aid pathologists and gynecologic oncologists in recognition and management. However, several issues limit clinical utility and reproducibility. Clarifying case presentation, deduplicating records, tightening methods/reporting, and correcting a few citation and language issues will substantially strengthen the paper.

#1. Present the index case with full pathology documentation (gross, histology, IHC).
At present, the “systematic review” dominates while the institutional case is not demonstrably shown. For a “case report and literature review”–type manuscript, please include:
-Pre-excision clinical photo or gross specimen image with a scale bar.
-Representative H&E at low and high power (highlighting villoglandular architecture, goblet/Paneth cells, intracytoplasmic mucin).
-A compact IHC panel figure (e.g., CK20, CDX2, CK7, p16, SATB2 if performed), with scoring/percentages in the legend.
This will anchor the review in a concrete, well-documented example and be highly useful to readers.

#2. Replace Table 3 with data that directly supports diagnostic practice.
As currently formatted, Table 3 is unlikely to be useful for most readers. Please substitute it with a table (or figure) summarizing the index case pathologic and immunophenotypic details (gross findings, key morphologic features, IHC profile, stage, treatment, and outcome). Consider adding a separate concise table that harmonizes IHC patterns across published VAIt cases (CK20/CDX2/SATB2/CK7/p16) to highlight consistencies and pitfalls.

#3. Resolve duplicate counting: Natsume (2024) and Fujii (2025) “Case 2” refer to the same patient. Please deduplicate these reports and correct aggregate counts (total cases, nodal positivity, stage distribution, outcomes). Update the PRISMA flow and all denominators accordingly, and indicate how duplicates were identified and handled.

#4. Correct miscitation at lines 333.
The in-text reference currently points to the wrong citation. It should reference [33] Fujii et al. Please amend the in-text citation and the reference list if needed.

#5. Please include a PRISMA flow diagram with numbers at each step (records identified, screened, excluded with reasons, included).

#6. Because VAIt can mimic metastatic lower GI adenocarcinoma, explicitly state how primary origin was adjudicated in each included case (clinical imaging, colonoscopy, anoscopy, GI work-up; absence of synchronous GI primary). If such work-ups were not uniformly reported, acknowledge as a limitation and consider a sensitivity analysis restricted to cases with documented exclusion of GI primaries.

#7. Please summarize the most reproducible IHC signature (e.g., CK20+/CDX2+ with variable CK7/p16) and comment on SATB2 where reported. If p16 is variably positive, clarify whether block-type/diffuse staining was observed (to avoid over-calling an HPV-related pattern).

#8. Improve grammatical precision: “clinic, pathologic, immunohistochemical characteristic” → “clinical, pathological, and immunohistochemical characteristics.”

Author Response

REVIEWER 3

Comment #0

  1. A) This manuscript addresses an exceptionally rare entity, intestinal-type vulvar adenocarcinoma (VAIt), by combining a systematic literature review with a single institutional case. The topic is clinically relevant, the rarity justifies aggregation, and the synthesis could aid pathologists and gynecologic oncologists in recognition and management. However, several issues limit clinical utility and reproducibility. Clarifying case presentation, deduplicating records, tightening methods/reporting, and correcting a few citation and language issues will substantially strengthen the paper.
  2. B) Response: We thank the Reviewer for the kind comment.
  3. C) Location: \

Comment #1

  1. A) Present the index case with full pathology documentation (gross, histology, IHC).
    At present, the “systematic review” dominates while the institutional case is not demonstrably shown. For a “case report and literature review”–type manuscript, please include:
    -Pre-excision clinical photo or gross specimen image with a scale bar.
    -Representative H&E at low and high power (highlighting villoglandular architecture, goblet/Paneth cells, intracytoplasmic mucin).
    -A compact IHC panel figure (e.g., CK20, CDX2, CK7, p16, SATB2 if performed), with scoring/percentages in the legend.
    This will anchor the review in a concrete, well-documented example and be highly useful to readers.
  2. B) Response: We thank the Reviewer for the comment. In fact, in this manuscript the “systematic review” dominates while the institutional case is not demonstrably shown because we designed this study as a systematic review and we decided to include the information about the case from Our Institution in Supplementary Material in accordance with the Editorial Board of the Journal. However, we are available to move such information to the main text if the Reviewer and the Editor decide to. We also included a novel table (Table 3) to compare IHC characteristics of the included cases.
  3. C) Location: Table 3

Comment #2

  1. A) Replace Table 3 with data that directly supports diagnostic practice.
    As currently formatted, Table 3 is unlikely to be useful for most readers. Please substitute it with a table (or figure) summarizing the index case pathologic and immunophenotypic details (gross findings, key morphologic features, IHC profile, stage, treatment, and outcome). Consider adding a separate concise table that harmonizes IHC patterns across published VAIt cases (CK20/CDX2/SATB2/CK7/p16) to highlight consistencies and pitfalls.
  2. B) Response: We thank the Reviewer for the comment. Information about gross findings, key morphologic features, IHC profile, stage, treatment, and outcome are reported in Tables 1 and 2. However, we agree to add a more concise table to harmonize IHC patterns, so we included a new Table 3 with such information.
  3. C) Location: Table 3

Comment #3

  1. A) Resolve duplicate counting: Natsume (2024) and Fujii (2025) “Case 2” refer to the same patient. Please deduplicate these reports and correct aggregate counts (total cases, nodal positivity, stage distribution, outcomes). Update the PRISMA flow and all denominators accordingly, and indicate how duplicates were identified and handled.
  2. B) Response: We thank the Reviewer for the comment. We corrected the duplication and the other related information in the results section.
  3. C) Location:

Comment #4

  1. A) Correct miscitation at lines 333.
    The in-text reference currently points to the wrong citation. It should reference [33] Fujii et al. Please amend the in-text citation and the reference list if needed.
  2. B) Response: We thank the Reviewer for the comment. We corrected the miscitation in the revised manuscript.
  3. C) Location: Tables

Comment #5

  1. A) Please include a PRISMA flow diagram with numbers at each step (records identified, screened, excluded with reasons, included).
  2. B) Response: We thank the Reviewer for the comment. We included the 2020 PRISMA flow diagram in the revised manuscript (Figure 1).
  3. C) Location: Figure 1

Comment #6

  1. A) Because VAIt can mimic metastatic lower GI adenocarcinoma, explicitly state how primary origin was adjudicated in each included case (clinical imaging, colonoscopy, anoscopy, GI work-up; absence of synchronous GI primary). If such work-ups were not uniformly reported, acknowledge as a limitation and consider a sensitivity analysis restricted to cases with documented exclusion of GI primaries.
  2. B) Response: We thank the Reviewer for the comment. Unfortunately, such information was not uniformly reported in the included studies, therefore we did not include it in our manuscript. We acknowledged this limitation in the revised manuscript.
  3. C) Location: 291-293

Comment #7

  1. A) Please summarize the most reproducible IHC signature (e.g., CK20+/CDX2+ with variable CK7/p16) and comment on SATB2 where reported. If p16 is variably positive, clarify whether block-type/diffuse staining was observed (to avoid over-calling an HPV-related pattern).
  2. B) Response: We thank the Reviewer for the comment. We added a more concise table to harmonize IHC patterns.
  3. C) Location: Table 3

Comment #8

  1. A) Improve grammatical precision: “clinic, pathologic, immunohistochemical characteristic” → “clinical, pathological, and immunohistochemical characteristics.”
  2. B) Response: We thank the Reviewer for the comment. We corrected grammatical precision the in the revised manuscript.
  3. C) Location: Throughout the manuscript

Reviewer 4 Report

Comments and Suggestions for Authors

 Some kinds of colorectal cancer are  currently successfuly treated  with immunotherapeutics, The authors only mention it. It is possible that some data from colonic cancer treatment may be transfered to VCiT therapy

Author Response

REVIEWER 4

Comment #1

  1. A) Some kinds of colorectal cancer are  currently successfuly treated  with immunotherapeutics, The authors only mention it. It is possible that some data from colonic cancer treatment may be transfered to VCiT therapy
  2. B) Response: We thank the Reviewer for the clever comment. We discussed a possible application of immunotherapy in VAIt based on the molecular similarities detected in colorectal cancer.
  3. C) Location: Lines 362-373